# The genetics and evolution of eye color in domestic pigeons (*Columba livia*)

Si Si[1,2ʘ], Xiao Xu[1,2ʘ]*, Yan Zhuang[1,2], Xiaodong Gao[3], Honghai Zhang[3], Zhengting Zou[4]*, Shu-Jin Luo[1,2]*

1 The State Key Laboratory of Protein and Plant Gene Research, School of Life Sciences, Peking University, Beijing, China, 2 Peking-Tsinghua Center for Life Sciences, Academy for Advanced Interdisciplinary Studies, Peking University, Beijing, China, 3 College of Life Sciences, Qufu Normal University, Qufu, Shandong, China, 4 Key Laboratory of Zoological Systematics and Evolution, Institute of Zoology, Chinese Academy of Sciences, Beijing, China

ʘ These authors contributed equally to this work.
* xusheau@126.com (XX); zouzhengting@ioz.ac.cn (ZZ); luo.shujin@pku.edu.cn (S-JL)

**Data Availability Statement:** The sequencing data for domestic pigeon accessions from this study have been deposited in the NCBI Sequence Read Archive (BioProject ID: PRJNA682513).

## Abstract

The eye color of birds, generally referring to the color of the iris, results from both pigmentation and structural coloration. Avian iris colors exhibit striking interspecific and intraspecific variations that correspond to unique evolutionary and ecological histories. Here, we identified the genetic basis of pearl (white) iris color in domestic pigeons (*Columba livia*) to explore the largely unknown genetic mechanism underlying the evolution of avian iris coloration. Using a genome-wide association study (GWAS) approach in 92 pigeons, we mapped the pearl iris trait to a 9 kb region containing the facilitative glucose transporter gene *SLC2A11B*. A nonsense mutation (W49X) leading to a premature stop codon in SLC2A11B was identified as the causal variant. Transcriptome analysis suggested that SLC2A11B loss of function may downregulate the xanthophore-differentiation gene *CSF1R* and the key pteridine biosynthesis gene *GCH1*, thus resulting in the pearl iris phenotype. Coalescence and phylogenetic analyses indicated that the mutation originated approximately 5,400 years ago, coinciding with the onset of pigeon domestication, while positive selection was likely associated with artificial breeding. Within Aves, potentially impaired SLC2A11B was found in six species from six distinct lineages, four of which associated with their signature brown or blue eyes and lack of pteridine. Analysis of vertebrate SLC2A11B orthologs revealed relaxed selection in the avian clade, consistent with the scenario that during and after avian divergence from the reptilian ancestor, the SLC2A11B-involved development of dermal chromatophores likely degenerated in the presence of feather coverage. Our findings provide new insight into the mechanism of avian iris color variations and the evolution of pigmentation in vertebrates.

## Author summary

Birds exhibit striking eye color variations, providing a unique angle for understanding avian evolution. Here we identified the genetic basis of the pearl (white) iris color in domestic pigeons (*Columba livia*) to a nonsense mutation W49X in *SLC2A11B* via whole

**Funding:** The project was supported by the National Natural Science Foundation of China (http://www.nsfc.gov.cn) (31970537 to X. X., and 32070598 to S. -J. L.), the National Key Research and Development Program of China (2017YFF0210303 to S. -J. L.), and the Peking-Tsinghua Center for Life Sciences (http://www.cls.edu.cn) (to S. -J. L.). The funders had no role in study design, data collection and analysis, decision to publish, or preparation of the manuscript.

**Competing interests:** The authors have declared that no competing interests exist.

genome sequencing and genome-wide association study (GWAS) approaches. *SLC2A11B* is a gene with known roles in fish pigment cells differentiation and transcriptome analysis indicated that SLC2A11B loss of function may downregulate the xanthophore-differentiation gene *CSF1R* and the key pteridine biosynthesis gene *GCH1*, resulting in the pigeon's pearl iris phenotype. The *SLC2A11B* variant was estimated to have originated at approximately 5,400 years ago coinciding with the onset of pigeon domestication and was then under positive selection likely associated with artificial breeding. Potentially impaired SLC2A11B was also found in six species from six distinct avian lineages. Analysis of vertebrate SLC2A11B orthologs revealed relaxed selection in the avian clade, consistent with the scenario that the SLC2A11B-involved development of dermal pigment cells likely degenerated in the presence of feather coverage. Our study sheds new light on the largely unknown genetic mechanism underlying the evolution of avian iris color variations.

## Introduction

Integumentary pigmentation plays essential roles in camouflage, sexual selection, communication, and thermoregulation in vertebrates [1–3]. The dynamic color change and diverse pigmentation of poikilothermic vertebrates are mostly attributed to neural crest-derived dermal chromatophores, which are generally divided into three main categories: xanthophores/erythrophores (yellow to red), iridophores/leucophores (iridescent color or white), and melanophores (black) [4–6].

The eye color of a bird, usually referring to the color of the iris, is derived from both pigmentation and structural coloration, that is, the presence of materials that diffract light. In homeothermic birds, with outer feather coverage masking skin pigmentation, dermal chromatophores may have undergone relaxed selection and become subject to evolutionary demise [7]. However, the avian iris maintains the potential for complete development of all types of pigment cells that are comparable to the chromatophores in poikilothermic vertebrates, probably due to its external, exposed location where chromatophores are under constant selective pressure, thus remaining as a "pigment cell refugium" during avian evolution [7].

The pigment cells are located in the eye stroma and the anterior layer of the iris consists of loose vascular connective tissue. Eye color in birds varies by the presence or absence of pigment cells in the iris and the content of pigments in those cells and exhibits striking interspecific variations, ranging from black and dark brown to brilliant colors that cover nearly the full spectrum of the rainbow [7–9]. Intraspecific eye color variation is also common and often associated with age and sex in wild birds [8,10–12]. Although the evolutionary drive shaping avian eye color remains largely unknown, recent studies have shed light on the possible coevolution of eye color and behavior or activity rhythm in birds [13–16]. Iris color variation may reflect unique evolutionary histories and ecological adaptions and thus provides a unique angle for understanding avian radiation, as well as pigmentation evolution across vertebrates.

The domestic pigeon (*Columba livia*) was derived from its synonymic wild ancestor, the rock pigeon, and its initial domestication is believed to have occurred approximately 5,000 years ago in the Mediterranean region [17–19]. After the onset of domestication, pigeons have been subject to intense selective breeding to have produced a wide array of phenotypic diversity within the species [17,18,20]. Recent studies have revealed the molecular basis of some intraspecific variations in domestic pigeons, such as plumage pigmentation, feather ornamentation, epidermal appendages, and navigation behavior, and highlighted the genes that might play important roles in avian evolution with respect to these features [21–29].

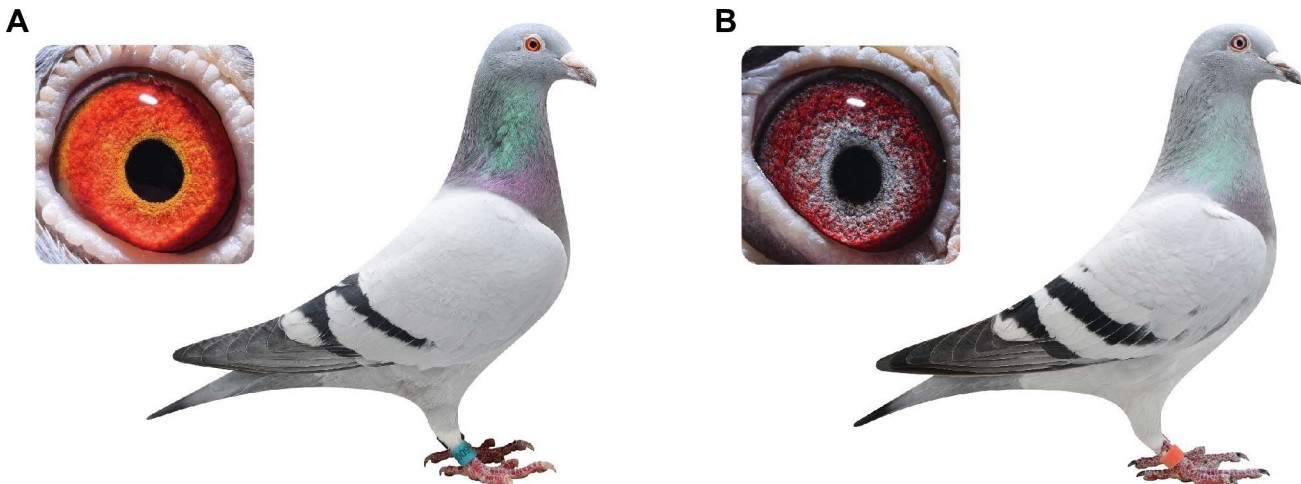

**Fig 1. Iris color variation in domestic pigeons.** (A) A pigeon with gravel eyes (wild type) exhibits a bright orange iris color; (B) A pigeon with pearl eyes exhibits a grayish-white iris color. The red background in both gravel and pearl irises is from capillaries. Photo credit: J. -O. Gao.

The domestic pigeon exhibits three major types of iris color: yellow to orange "gravel" (wild type, Fig 1A), white "pearl" (Fig 1B), and black "bull" eyes [7–9]. The gravel and pearl irises in pigeons contain bright pigment cells with birefringent crystals in the anterior stromal tissue, while the bull eye results from a complete absence of stromal pigment cells [30]. Crystalline guanine was identified as the major pigment in both gravel and pearl irises, and at least two yellow fluorescing pteridine materials were identified leading to the yellow color tone in the gravel iris [30]. Therefore, the stromal pigment cell in the gravel iris is also referred to as a "reflecting xanthophore" [30]. By contrast, the white color of a pearl eye is the result of the presence of guanine only without the manufacture of yellow pigments (pteridines) in the pigment cells. All red colors in gravel and pearl eyes are from the blood vessels in the outer iris (Fig 1). The irises of all newly hatched pigeons are bull-eyed dark, and those with gravel or pearl eyes gradually develop a brighter color after two to three months. Empirical breeding records indicate that the pearl eye color is an autosomal recessive trait relative to the gravel eye, denoted as the *Tr* locus, and that the bull eye trait is associated with white feathers [31]. Similar stromal pigment cells and/or developmental color changes were found in the irises of other avian species [8,12,32, 33], suggesting that the domestic pigeon is an ideal model to study the evolution of avian pigmentation variations.

Here, we investigated the genetic basis of pearl irises in domestic pigeons. We identified the genetic mutation in SLC2A11B responsible for the iris color change from gravel to pearl and found that the pearl iris trait likely originated approximately 5,400 years ago as a domesticated trait under artificial selection. Evolutionary analyses revealed a conserved role for SLC2A11B in avian eye color and associated its mutation with a lack of yellow iris pigmentation in various lineages of birds. Analysis of vertebrate SLC2A11B orthologs further revealed a relaxation of selection in the avian clade after its divergence from the reptilian ancestor, shedding new light on the origin and evolution of avian coloration.

## Results

### The *Tr* locus is mapped to a 9 kb genomic region

A case-control genome-wide association study (GWAS) based on whole-genome resequencing was performed in domestic pigeons (racing homers) to identify the genomic region

responsible for pearl eyes. Whole-genome sequencing data were generated from 49 gravel-eyed and 43 pearl-eyed individuals at approximately 10× genome coverage each (S1 Table). Of the 28,628,324 SNPs initially identified, 2,490,056 passed genotype filtering and quality control and were used in subsequent analysis (S1 Fig). Significant signals of association ($P < 2.01 \times 10^{-8}$, after Bonferroni correction) were observed in 160 consecutive SNPs from the pigeon genome (Cliv_2.1, GenBank accession: GCA_000337935.2) [34] scaffold AKCR02000030.1 (hereafter referred to as scaffold 30), which spanned a single 341 kb region associated with the pearl-eyed phenotype (minimum $P = 1.07 \times 10^{-16}$) (Figs 2A and S2). No SNPs from other scaffolds showed a significant signal of association with the iris phenotypes (Figs 2A and S2). To fine-map the pearl eye *Tr* locus, haplotypes of the mapped region on scaffold 30 were built for each individual based on whole-genome sequencing data. While wild-type individuals exhibited multiple haplotypes in this region, a single 9 kb haplotype (scaffold 30: 1894006–1903342) was shared by all 43 pearl-eyed pigeons, highlighting this region as a strong candidate for the *Tr* locus (Fig 2B).

## A premature stop codon in SLC2A11B causes the pearl eye color in pigeon

The mapped 9 kb interval of the *Tr* locus contained only one protein-coding gene according to the gene annotation of the reference genome (Cliv_2.1): A306_00005299 (Fig 2B). A306_00005299 was annotated as a 12-transmembrane helix transporter gene, solute carrier family 2 member 11-like (S3A and S3B Fig), and was further identified as *SLC2A11B* (solute carrier family 2, facilitative glucose transporter, member 11b) based on gene homology and synteny in vertebrates (avian, reptile, and fish). The full-length pigeon *SLC2A11B* transcript was validated by RT-PCR from the iris tissue RNA of a gravel-eyed pigeon. *SLC2A11B* is known to play an essential role in the differentiation of leucophores and xanthophores in medaka fish [35] and hence is a strong candidate for pearl eye color in domestic pigeons. To identify the causative mutation, we screened variations in the pigeon SLC2A11B coding region and identified a G-to-A transition in exon 3 that introduced a premature stop codon (W49X) to truncate the 448 downstream amino acids (Fig 3A). SLC2A11B loss of function in medaka fish prevented the development of xanthophores and eliminated the yellow pigment in leuco-phores at embryonic/larval stages [35]. SLC2A11B-knockout zebrafish also exhibited reduced yellow pigment due to defects in xanthophore differentiation (https://zmp.buschlab.org/gene/ENSDARG00000093395). The pigmentation change of the iris stromal pigment cells in pearl-eyed pigeons resembles the leucophore color switch from yellow to white in medaka fish larva with SLC2A11B loss-of-function [35], thus supporting W49X of SLC2A11B as the causal mutation responsible for the pearl eye color. This nonsense mutation was further validated in 146 gravel-eyed and 146 pearl-eyed individuals by PCR and Sanger sequencing. All gravel-eyed pigeons carried at least one wild-type allele, while 141 out of the 146 pearl-eyed individuals were homozygous for the mutant allele, consistent with its recessive mode of inheritance. The only exception included five pearl-eyed pigeons that were heterozygous for the *SLC2A11B* W49X variant (Fig 3B) and had no other mutations found in the exons of *SLC2A11B*. Such phenotype-genotype incompatibility in these five outliers could have been due to mis-phenotyping of young pigeons or observer error.

## Genes with known functions in xanthophore/leucophore pigmentation are downregulated in SLC2A11B W49X pigeon iris

The expression pattern of *SLC2A11B* was profiled in multiple tissues from nine wild-type domestic pigeons. *SLC2A11B* expression was equal and extremely low in both skin and feather buds ($P = 0.736$) (Fig 4A), consistent with the absence of the non-melanocyte pigment cells in

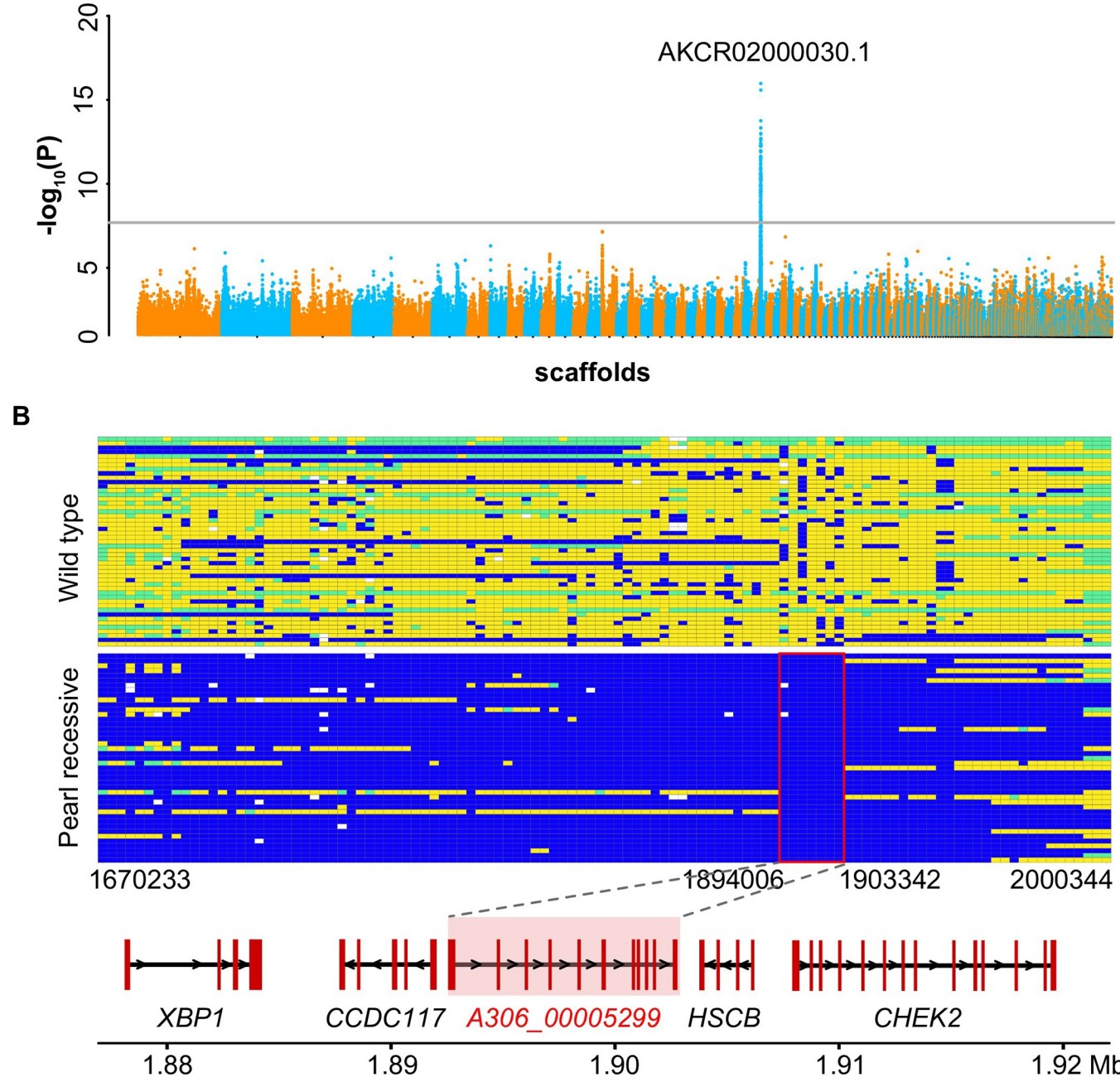

**Fig 2. Genomic mapping of the *Tr* locus.** (A) The Manhattan plot of the GWAS analysis for the *Tr* locus. The gray line indicates the Bonferroni-corrected critical value ($P = 2.01 \times 10^{-8}$). The scaffold with the strongest association signal is marked. (B) The genotypes of 92 pigeons across the mapped region associated with the pearl eye. Each row represents one individual, and each column represents an SNP/indel. Blue indicates homozygosity for the major allele in pearl-eyed pigeons, yellow indicates homozygosity for the minor allele in pearl-eyed pigeons, green indicates heterozygosity, and white indicates missing data. A shared haplotype (scaffold 30: 1,894,006–1,903,342) was identified. Genes located in and around the shared region are labeled.

avian skin. On the contrary, *SLC2A11B* was highly expressed in iris, exhibiting an 8.88-fold increase ($P < 0.001$) relative to that of the skin tissue (Fig 4A). Using the expression in skin as a baseline, higher level of expression of *SLC2A11B* was also observed in several other tissues,

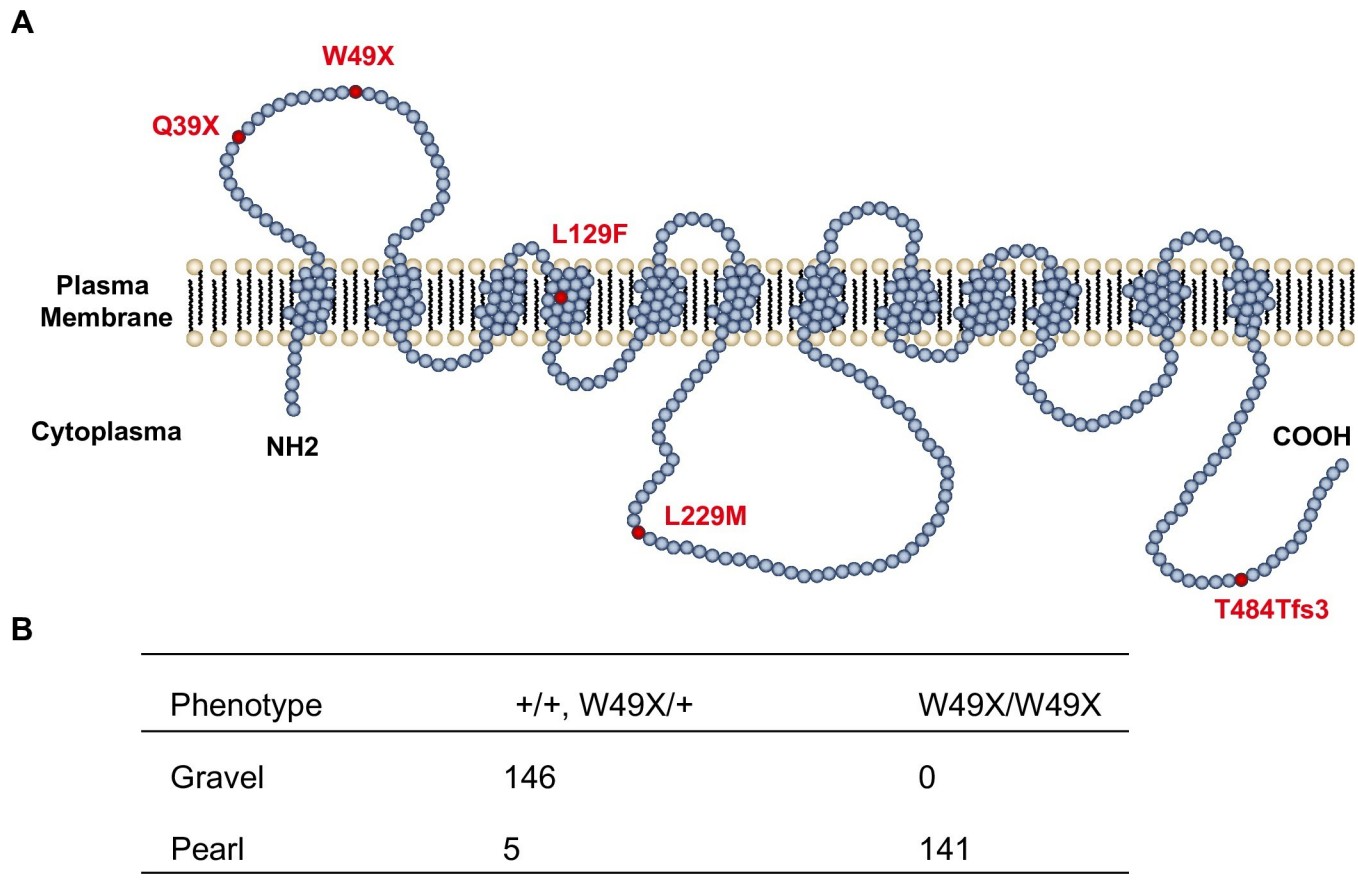

**Fig 3. SLC2A11B mutations in avian species.** (A) Schematic diagram of the SLC2A11B protein with 12 transmembrane domains. The SLC2A11B mutations in birds are highlighted in red. (B) Validation of the SLC2A11B W49X mutation in domestic pigeons.

including the muscle (10.85-fold change, $P < 0.001$), retina (8.35-fold change, $P < 0.001$), brain (4.27-fold change, $P < 0.001$), heart (2.44-fold change, $P < 0.001$), and liver (1.69-fold change, $P = 0.022$) (Fig 4A). These patterns jointly suggest the role of SLC2A11B in the pigmentation process of pigeons may be restricted to iris only among the epithelial tissues, while exerting pleiotropic effects on some internal organs.

Transcriptome analysis of iris tissues from four gravel-eyed and five pearl-eyed pigeons (S2 Table) identified 337 differentially expressed genes (DEGs) (FDR < 0.05; Figs 4B and S4 and S5, S3 Table). *SLC2A11B* was not significantly differentially expressed, but a downregulation trend of *SLC2A11B*, likely caused by nonsense-mediated RNA decay, was evident in pearl-eyed individuals (Fig 4B), which was further confirmed by qPCR (Fig 4C). Among these DEGs, we examined the genes involved in pteridine biosynthesis or xanthophore/leucophore differentiation. We identified that GTP cyclohydrolase 1 (*GCH1*), the gene encoding a rate-limiting enzyme in the pteridine pathway, was significantly downregulated (3.6-fold) in the iris tissues of pearl-eyed pigeons (Fig 4B), consistent with the absence of yellow pigment. We also found that colony-stimulating factor 1 receptor (*CSF1R*), a gene encoding a receptor tyrosine kinase required for the differentiation of the pteridine-containing xanthophore [36–38], was significantly downregulated (1.8-fold) in pearl irises (Fig 4B). The reduced expression of *GCH1* and *CSF1R* was further confirmed by qPCR (Fig 4C). Overall, these differential transcriptome profiles between gravel and pearl eyes in the pigeon suggested that the loss-of-function variant of SLC2A11B may affect the differentiation of xanthophore-like stromal pigment

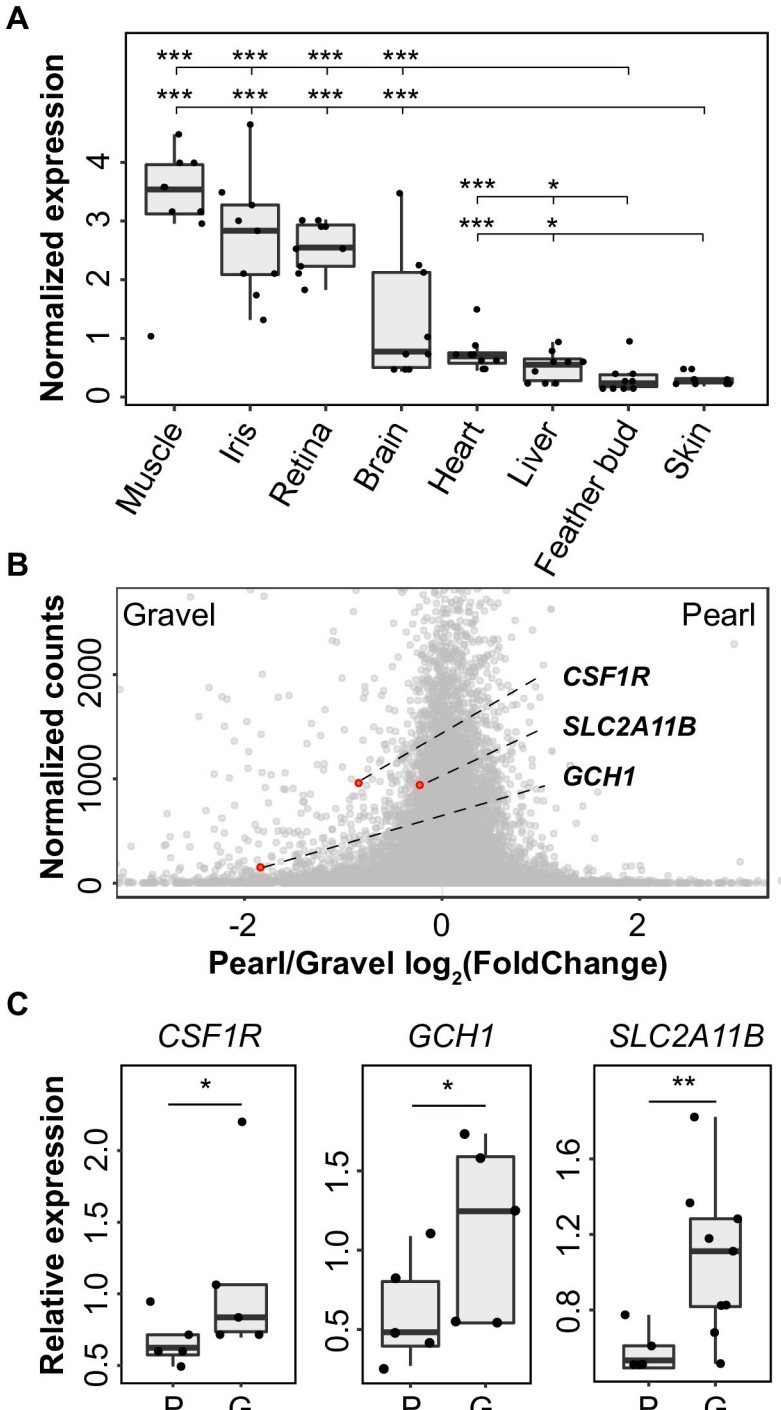

**Fig 4. The expression profile of *SLC2A11B* in pigeon tissues and genes with differential expression between pearl and gravel irises.** (A) Normalized expression of *SLC2A11B* in various tissues from wild-type pigeons as determined by qPCR analysis. Boxes span the first to third quartiles, bars extend to the minimum and maximum observed values, black lines indicate the medians, and circles represent the data points. (B) Normalized comparison of *SLC2A11B* expression between the pearl and gravel irises by transcriptome sequencing. Differentially expressed genes involved in xanthophore differentiation or pteridine synthesis are highlighted. (C) qPCR validation of differentially expressed genes (*CSF1R*, *GCH1*, and *SLC2A11B*) between pearl and gravel irises. $^*P < 0.05$, $^{**}P < 0.01$, $^{***}P < 0.001$.

cells and pteridine biosynthesis machinery, resulting in the reduction of yellow pigment and hence the pearl iris color in pigeon.

In addition, located in the same linkage group with *SLC2A11B* we found six DEGs (*IGLL1*, *GGT1*, *TBX3*, *KNTC1*, *PRC1*, and *A306_00008692*; S3 Table), among which *GGT1* and *TBX3* are involved in the regulation of melanocyte in mammals [39,40]. Although we cannot completely rule out the possibility that potential regulatory variants within the GWAS candidate interval may cause pearl eyes, this alternative hypothesis is highly unlikely, as none of the six DEGs is known to have any function associated with xanthophore or leucophore, and hence could not explain the pigmentation change in pearl-eyed pigeons.

## Pearl eye is a domesticated trait under artificial selection in pigeons

We surveyed the SLC2A11B mutation in 45 pigeons that represented 35 breeds worldwide [21] to investigate the evolution and origin of the pearl-eyed trait in pigeons (S4 Table). The variant was found in 29 of the 45 pigeon genomes, or 20 of the 35 breeds (S4 Table). Such widespread presence of SLC2A11B W49X allele is consistent with the phenotypic prevalence in pigeons, in that at least 17 of the 35 breeds are independently documented to carry the pearl-eyed trait (S4 Table).

To further investigate the origin of the SLC2A11B W49X mutation, a neighbor-joining phylogenetic tree was reconstructed based upon the 9 kb nonrecombining *Tr* region from 139 domestic pigeons (35 fancy pigeons, 2 feral pigeons, and 102 racing homers) and one *C. rupestris* that is considered sister taxa to all domestic pigeons (Fig 5A). All haplotypes containing the W49X mutation ($tr^{mut}$) formed a monophyletic group nested within the wild-type ($Tr^+$) clade (Figs 5A and S6), indicating a derived state of the SLC2A11B variants originating from the wild type via one single mutation event. The derived status of $tr^{mut}$ in relation to $Tr^+$ was also evident in the median-joining haplotype network (S6 Fig). In addition, the fact that all mutant haplotypes cluster together in the gene tree but spread across different breed sources and geographic regions (Fig 5A) supports that the mutation is old and likely occurred prior to the establishment of modern pigeon breeds. We further estimated the time to the most recent common ancestor (TMRCA) for all *SLC2A11B* haplotypes bearing the W49X variant at approximately 5,400 years ago (95% CI: 4,700–5800 years ago) (Fig 5B), a time period coinciding with pigeon domestication that was estimated at more than 5,000 years ago.

Signals of selection for the pearl-eyed trait in pigeons were tested based on 139 genome datasets through evaluation of integrated haplotype score (iHS), nucleotide diversity ($\pi$), Tajima's D, and extended haplotype homozygosity (EHH). We first applied iHS analysis to scaffold 30, where the *Tr* locus is located, and identified continuous signals of positive selection at the *Tr* locus and its adjacent region (scaffold 30: 1.4–2.0 Mb) (Figs 6A and S7A and S7B). Decreased nucleotide diversity and negative Tajima's D were evident across the SLC2A11B haplotypes with the W49X mutant (Fig 6C and 6D), consistent with a selective sweep scenario. The mutant haplotypes exhibited longer homozygosity than the wild-type haplotypes in the EHH analysis (Figs 6B and S7C and S7D), in support of the occurrence of selection for the variant. These evidences jointly illustrated that the prevalence of the pearl iris trait in pigeons was likely a result of artificial selection during pigeon domestication.

## SLC2A11B is associated with avian iris color variation

Given the important role of SLC2A11B in determining the iris color variations in domestic pigeons, it is plausible that defects in SLC2A11B in other bird species may also be responsible for the lack of pteridine and hence contributing to the avian iris color diversity. We screened sequence variations across the *SLC2A11B* coding region in 34 avian species, whose iris

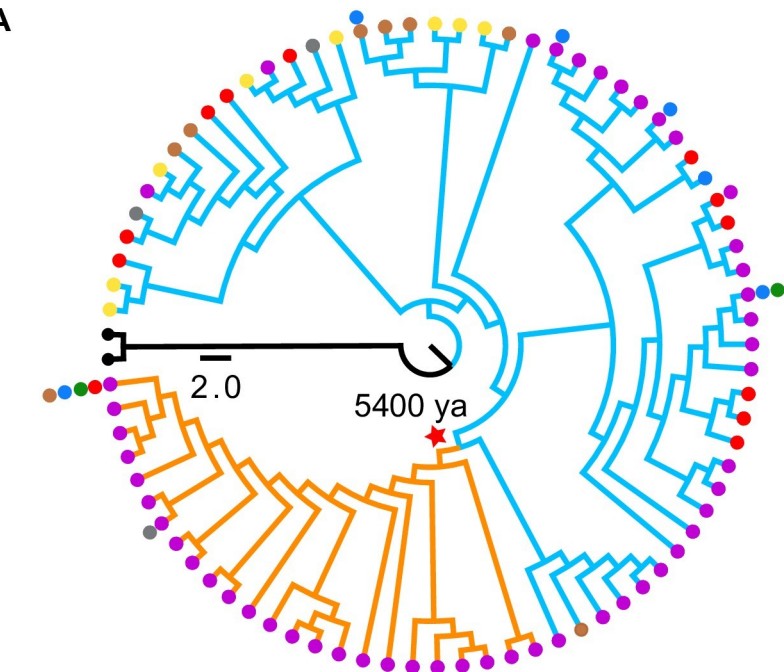

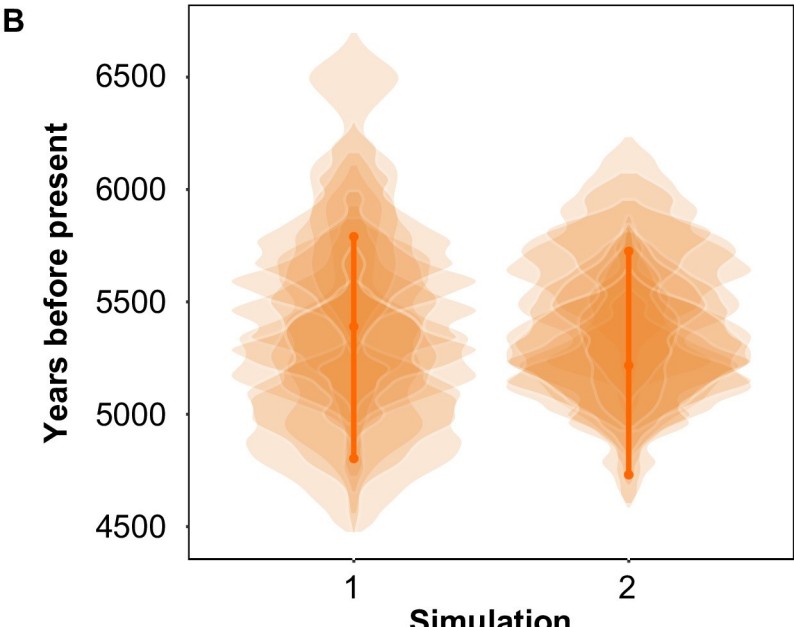

**Fig 5. The origin of the pearl iris variant $tr^{mut}$ in domestic pigeons.** (A) Neighbor-joining (NJ) trees of the 9 kb nonrecombining *Tr* region based on 81 haplotypes from 140 unrelated individuals comprising 139 domestic pigeons (35 fancy pigeons, two feral pigeons, and 102 racing pigeons) and one *Columba rupestris* (outgroup). All nodes had more than 60% support from 1,000 bootstrap replicates. Orange branches indicate $tr^{mut}$ haplotypes, blue branches indicate $Tr^+$ haplotypes, and the black branch indicates the outgroup. Different circle colors represent the seven traditional breeds of domestic pigeons and the outgroup. The red star indicates the time of origin of the *Tr* mutation.

(B) Violin plots of the posterior distribution of the estimated TMRCA for the *Tr* locus in domestic pigeons. The generation time was set at one year. Two independent simulations were performed using the estimated recombination rates in 10 kb and 50 kb windows. In every simulation, 10 replicate MCMCs were plotted with transparency.

pigment information [9,41] and genomes data are both available (S5 Table), and identified 335 non-synonymous variations. Four amino acid changes, including one nonsense (Q39X), one reading frameshift (T484Tfs3), and two missense (L129F and L229M) mutations (Figs 3A and S8 and S5 Tables), are predicted to impair SLC2A11B function in six bird species.

A nonsense mutation Q39X is identified in the house sparrow (*Passer domesticus*), whose eyes are sepia brown without pteridines in the iris tissue. Q39X is predicted to cause SLC2A11B loss-of-function by truncating the downstream 458 amino acid residues, and is consistent with the lack of yellow pigmentation in the iris of house sparrows.

T484Tfs3 is found in the American anhinga (*Anhinga anhinga*), double-crested cormorant (*Phalacrocorax auritus*), and greater rhea (*Rhea americana*). The reading frameshift mutation truncates the SLC2A11B C-terminus by 12 amino acids, most of which are evolutionarily conserved sites. The deleterious impact of T484Tfs3 on SLC2A11B is consistent with the pteridine-free iris of the double-crested cormorant (blue eye) [7] and greater rhea (dark brown eye) [41], though not in the American anhinga whose pale yellow iris contains pteridines [41].

The missense mutations L129F and L229M are detected in the grey parrot (*Psittacus erithacus*, white eye) and the Muscovy duck (*Cairina moschata*, brown eye) respectively. Both variants occur at the evolutionarily conserved sites (S8 Fig) and are predicted to affect SLC2A11B function (with a deleterious SIFT score less than 0.05). L229M is likely associated with the brown iris color or absence of pteridine in the Muscovy duck [41], whereas a direct link is not apparent for L129F, as the grey parrot shows colorless pteridines in its iris [41]. Nevertheless, given the multifactorial nature of pigmentation process, the genotype-phenotype association across evolutionarily divergent lineages of birds supports that SLC2A11B is involved in avian iris color diversity.

## SLC2A11B was under relaxed selection during or after avian divergence from its reptilian ancestor

With the development of full feather coverage in birds and/or their feathered non-avian dinosaur ancestors, the functional constraints on skin coloration may have been gradually lifted, followed by the degeneration of dermal chromatophores, leaving only those epidermal melanocytes responsible for feather pigmentation. It is likely that genes specialized in the development of xanthophores or iridophores, in this case *SLC2A11B*, could have experienced relaxation in natural selection during or after avian divergence from the ancestral reptile lineage.

To test this hypothesis, we analyzed SLC2A11B orthologs from 41 species, including five major vertebrate clades: Aves (birds), Crocodilia (crocodiles), Testudines (turtles), Squamata (scaled reptiles), and Teleostei (ray-finned fishes). RELAX analysis [42] was conducted, in which a statistic $k$ is calculated denoting the strength of selection on a "Test" set of branches normalized to that on a "Reference" branch set. As the test involves modeling and inference of multiple mutational and selection parameters, information provided by sequence states of a single gene (636 codon sites for *SLC2A11B* and 572 codon sites for *TYR*) on a few branches may bear limitation. To avoid spurious significance and to strengthen data inference, we set three different schemes of test/reference branch designation, involving only basal ("basal")/all internal ("internal")/all ("clade") branches of each group (Fig 7). For each scheme, relaxation of selection is tested for the branch(es) of each species group against the corresponding branches of the other four groups. We expected that birds show relaxation ($k < 1$) in the test

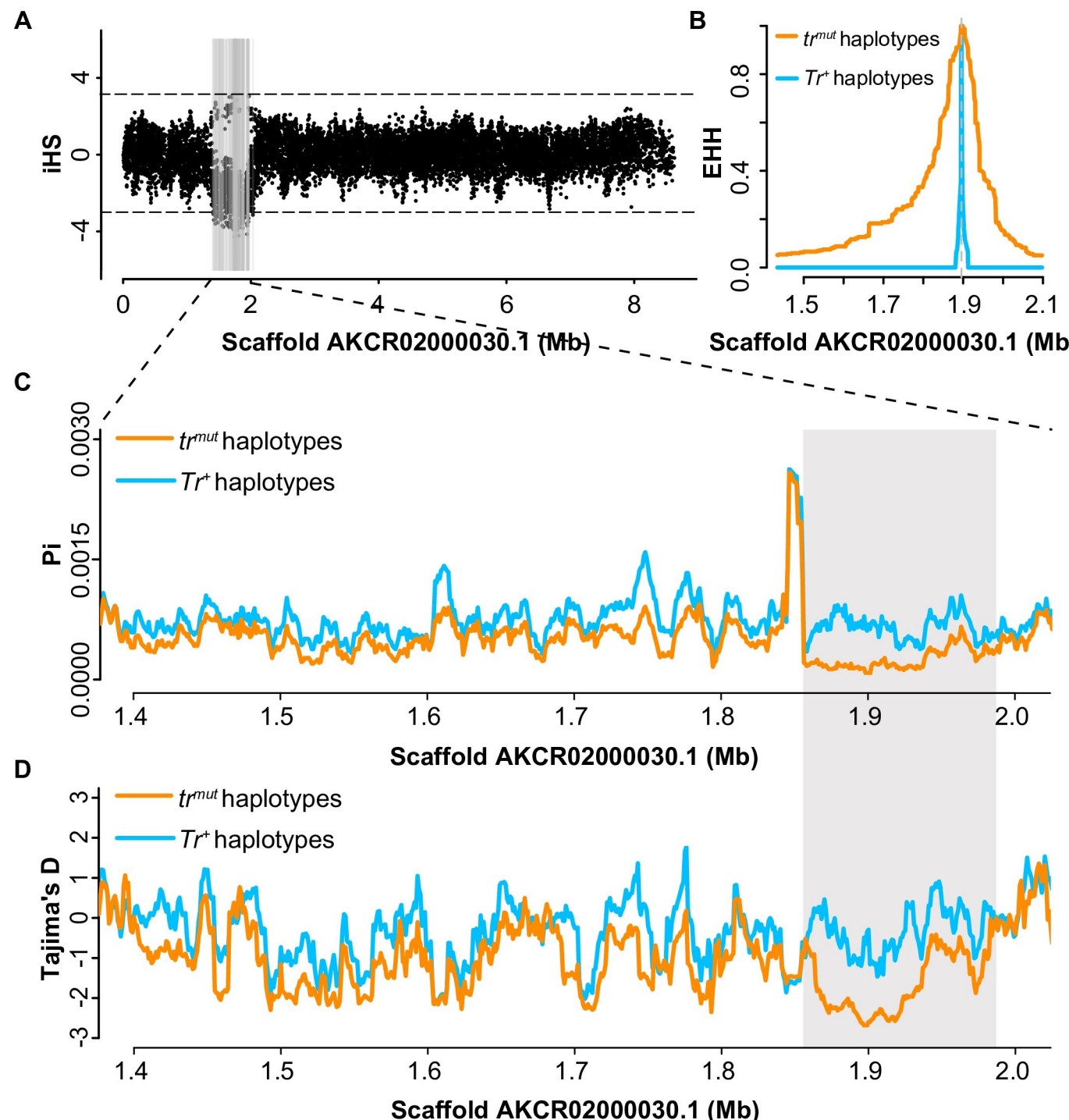

**Fig 6. Selection of the pearl iris variant $tr^{mut}$ in domestic pigeons.** (A) The distribution of integrated haplotype score (iHS) values on scaffold AKCR02000030.1. The gray lines indicate the significance of the absolute iHS scores of 3 or greater. (B) Extended haplotype homozygosity (EHH) decay across the $Tr$ locus region. Nucleotide diversity ($\pi$, Pi) (C) and Tajima's D (D) were calculated in 10 kb windows with a 1 kb step size for $tr^{mut}$ and $Tr^+$ haplotypes.

branches for *SLC2A11B*, while no such significance was expected for *TYR*. Consistent results across three different schemes, or, the same conclusions based on information from different branches, would be most convincing.

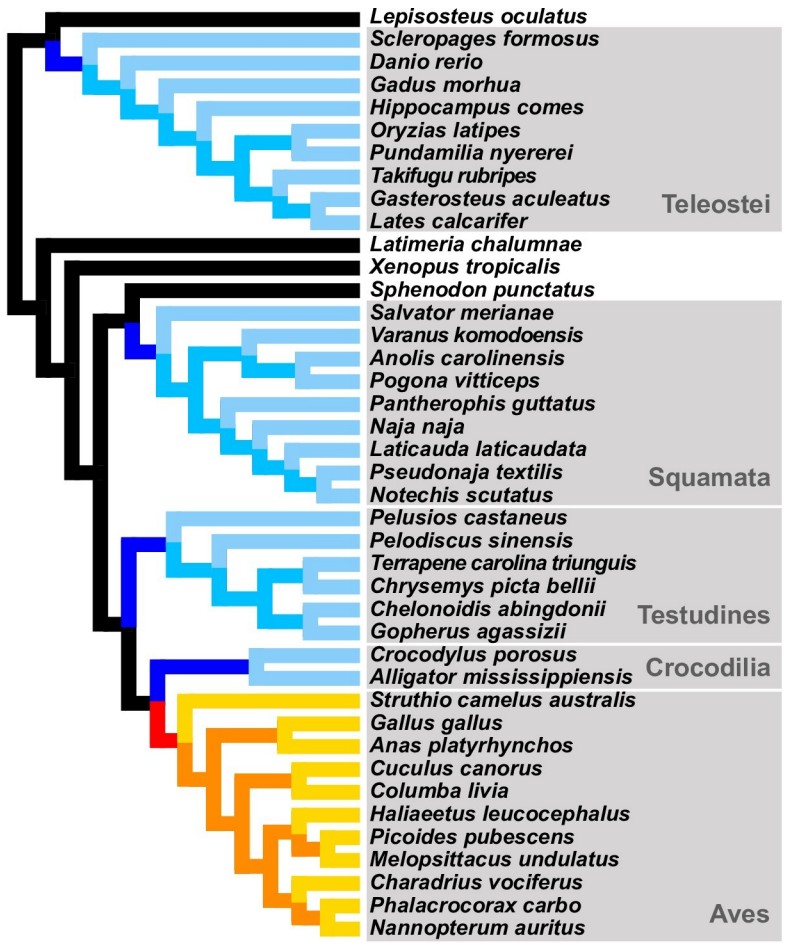

| Scheme | SLC2A11B | | TYR | |
|---|---|---|---|---|
| | *k* | corrected *P*-value | *k* | corrected *P*-value |
| Basal <br> ▬ vs. ▬ | **0.74** | 2E-03 | 0.57 | 1.0 |
| Internal <br> ⊏ vs. ⊏ | **0.10** | 2E-11 | 0.56 | 0.6 |
| Clade <br> ⊏ vs. ⊏ | **0.40** | <1E-300 | **0.38** | 5E-09 |

**Fig 7. RELAX tests indicate consistent relaxation of selection in avian *SLC2A11B* but not *TYR*.** Three different schemes of test/reference branch designations were illustrated by color-labeled branches (red/dark blue for "basal", red and orange/dark and intermediate blue for "internal", all warm colors/all cool colors for "clade") in the phylogeny and were tested separately (see Materials and Methods). The results with birds (warm-colored branches) as test branches are shown in the table below (see S6 Table for more results). The *k* values significantly smaller than 1, which indicate relaxation of selection, are highlighted in bold green font. The *P*-values shown are after Bonferroni correction for multiple testing.

Relaxed selection in SLC2A11B was consistently detected along the avian clade (k < 1) under all three schemes of test/reference branch designations, but not in turtles or teleosts under any setting (Fig 7 and S6 Table). Only a weak relaxation signal was partially detected in crocodilians under the "basal" scheme and in squamates under the "clade" scheme, yet neither was consistent across all schemes (S6 Table). In contrast, no relaxed selection was consistently supported for any clade by more than one schemes for *TYR*, a key melanogenesis gene that is expected to be evolutionarily conserved and under similar selective pressure across all vertebrates (Fig 7 and S6 Table). Specifically, test for avian *TYR* is only significant under the "clade" scheme but not significant under either "basal" or "internal" scheme. This indicates that the relaxation of selection in *TYR*, if any, did not arise until the relatively recent diversification of birds and hence is not likely related to the initial avian divergence from reptile ancestors. Overall, relaxation of selection in *SLC2A11B* in the avian clade is supported, consistent with the scenario that *SLC2A11B*-involved development of dermal chromatophores in modern birds likely went through a process of degeneration following the emergence of feather coverage.

## Discussion

Through whole-genome sequencing and GWAS, we identified a nonsense mutation W49X in SLC2A11B as most likely responsible for the pigmentation change in the iris of pearl-eyed domestic pigeons. Most of the pearl-eyed individuals tested (141 out of 146) were homozygous for this mutation, consistent with its recessive mode of inheritance. One possible explanation for the exceptions of five pearl-eyed heterozygous pigeons might be additional variation(s) in SLC2A11B that give rise to the same phenotype. Since no other mutation was identified from *SLC2A11B* coding regions, the additional mutation(s) involved in pearl iris color might be located in a different gene or in the regulatory elements of *SLC2A11B*. Alternatively, the phenotype-genotype discordance in these five pearl-eyed individuals carrying one wild-type *SLC2A11B* allele might be due to mis-phenotyping. The iris of newly hatched domestic pigeons is black and gradually switches to brighter colors after two to three months. Phenotypes of all the pigeons sampled in this study were recorded at approximately five months old, when the iris color was usually stable. However, it is possible that the five heterozygous individuals might be younger than the others or that there could be individual variance in the developmental stage of pigeons at which iris pigmentation change may still be in progress. Unfortunately, we were unable to trace back to these five discordant pigeons to confirm the potential phenotyping error, as they were lost during the homing championship in the same year.

The W49X mutation leads to the truncation of approximately 90% of the amino acids of SLC2A11B and is predicted to cause a total loss of function in the protein. Besides iris, high expression of *SLC2A11B* was evident in brain, muscle, and retinal tissues, suggesting potentially multiple roles of this protein in addition to its involvement in pigmentation pathways. Therefore, it would be reasonable to expect consequences of the mutation other than iris pigmentation change. However, it seems that the influence of the *SLC2A11B* mutation in pigeons was restricted to iris pigmentation, as no other abnormality was apparent in pearl-eyed individuals. This is an observation consistent with the report that the loss of function of SLC2A11B in fishes affected only pigmentation cells [35]. It is likely that the roles of SLC2A11B in non-pigmented cells of pearl-eyed pigeons (as well as other SLC2A11B mutants) could be compensated by the expression of other genes, most likely the other family members of solute carrier family 2.

Although the stromal pigment cells in the pigeon with gravel eyes are generally considered xanthophore-like, they show a reflecting effect that is normally absent in typical xanthophores. The stromal pigment cells in pigeons resemble the leucophores in medaka fish, in which the

orange pigment is present at the larval stage but diminishes in the SLC2A11B-defect mutant [35]. Therefore, it might be more appropriate to refer to pigeon iris stromal pigment cells as leucophore-like. Significantly reduced expression of *CSF1R*, a gene involved in xanthophore differentiation [36–38], was detected in the pearl iris (*SLC2A11B* mutant), which is consistent with the scenario that *CSF1R* is downstream of *SLC2A11B* in the regulatory network of xanthophore/leucophore differentiation. In addition, *GCH1*, the gene encoding the first and rate-limiting enzyme in the pteridine biosynthesis pathway [43,44], was also downregulated in the pearl iris. Since *GCH1* directly participates in pteridine synthesis, it is plausible that *GCH1* acts at the far end of the regulatory pathway of xanthophore/leucophore differentiation and triggers pteridine production upon receiving the upstream signal.

While the precise timing of pigeon domestication is unclear, it is commonly accepted that the pigeon was first domesticated at least 5,000 years ago in the Fertile Crescent [17–19] and probably served as a food resource for humans as early as 10,000 years ago [17,19,45]. The estimated time of the origin of the *SLC2A11B* W49X mutation at approximately 5,400 years ago is in line with the beginning of pigeon domestication, supporting the idea that pearl iris in domestic pigeons was a derived trait closely associated with the domestication process. Strong signals of positive selection were also detected, providing further evidence that the pearl iris trait has undergone artificial selection.

Even after thousands of years of domestication, little morphological difference exists today between a wild-type domestic pigeon and its conspecific rock pigeon; thus, it is reasonable to postulate that the pearl iris could serve as a marker for distinguishing a domesticated individual from its wild counterpart during the early stage of domestication. In modern breeds, the reason for the selection of the pearl iris is more complicated. As the eye is one of the classic judging criteria for pigeons, the pearl iris has been preferred in many show breeds, such as Show Homer, Runt, and Trumpeter. It is interesting that a very high frequency of pearl irises is observed in performance breeds as well, such as Tumbler, Roller, and Highflier. In addition to a random founder effect, breeders tend to hold an unfounded belief that pearl eyes in a pigeon are somewhat associated with intelligence and hence better performance.

We examined evolutionarily diverging bird species, whose iris pigmentation phenotype and genome data are both available, to explore the role of SLC2A11B in eye color determination. We found four mutations that might affect the SLC2A11B function in six bird species, within which Q39X in the house sparrow and L229M in the Muscovy duck well correspond to the lack of pteridines in their irises. The frameshift mutation T484Tfs3 is predicted to impair the function of SLC2A11B and is associated with the pteridine-absent iris color of the double-crested cormorant and greater rhea, though not with the American anhinga that carries the same mutation but has pale yellow eyes. This contradiction indicates that there could be other factors involved in the SLC2A11B pathway leading to pteridine production. Another inconsistency is L129F, which is also predicted to cause SLC2A11B deficiency, is present in the white-eyed grey parrot with pteridines in its iris tissue. It is interesting though the pteridines in the iris of grey parrot are colorless, other than ordinary yellow pigments. Whether such types of pteridines in the grey parrot are the results of SLC2A11B defect by L129F would be worth further investigation. Overall, despite the incongruence, association is evident between loss-of-function SLC2A11B variants and the absence of pteridines from iris in various avian taxa, thus supporting that the SLC2A11B pathway may serve as a common mechanism underlying the eye color diversity in Aves.

One landmark evolutionary transition setting the stage for the origin of birds was the development of feathery coverage that concealed their skin pigmentation in their feathered, non-avian dinosaur ancestors. The presence of fossilized non-avian dinosaurs with filaments or feathers suggested a complex integumentary covering formed prior to the avian divergence

[46–49]. Therefore, the dermal chromatophores underneath feather might already become functionally redundant and likely subject to evolutionary demise before the onset of birds. In modern birds, the epidermal melanocyte is the only pigment cell distributed throughout the body and plays an important role in integument pigmentation. One exception lies in the iris, which seems to maintain the full developmental potential for all types of chromatophores and therefore has been proposed to be a pigment cell "refugium" [7].

In this context, our findings provide new insights into the mechanism underlying the evolutionary changes in pigment cells in Aves. First, *SLC2A11B*, a gene first found to be involved in the differentiation of xanthophores and leucophores in fish, is absent in mammals but intact in birds. Among the pigeon epithelial tissues, the expression of SLC2A11B is specific to the iris. Evolutionary analysis of *SLC2A11B* orthologs in vertebrates supported that, relative to other reptiles and fishes, *SLC2A11B* was under relaxed selection in the avian clade. We proposed that during and after avian divergence from the reptilian ancestor, with the newly evolved feather coverage lifting the selection pressure on the dermal coloration beneath, the expression of *SLC2A11B* and other specialized pigmentation genes may have gradually switched down or off in the dermis, thus resulting in the degeneration of the dermal chromatophores in the avian ancestor. In the iris, however, the expression of *SLC2A11B* and other pigmentation genes remained functional, and the pigment cells were sustained.

The roles of iris pigmentation genes could have gone through a dynamic and complicated process during avian evolution. At the early stage of avian radiation, it is likely that the iris color of the ancient birds exhibited limited diversity, as the evolutionary constraints on the genes involved in the development of the non-melanophore chromatophores were just being released. Subsequently, mutations accumulated in the dermal pigmentation genes, and iris color diversity in birds likely emerged, some of which might be of adaptive significance and subject to selection along different avian lineages. Further studies on *SLC2A11B*, as well as other pigmentation genes in birds, promise to illuminate the adaptive, functional, and evolutionary processes involved in avian coloration.

## Materials and methods

### Ethics statement

All handling of animals and experimental protocols were approved by the Institutional Animal Care and Use Committee of Peking University (IACUC# LSC-LuoSJ-2) and the methods were performed in accordance with the relevant guidelines.

### Sample collection

Feather samples of domestic pigeons were collected at the Xiangguan Pigeon Racing Club, Panjin City, Liaoning Province, China, in 2017. This pigeon racing club housed over 7,000 newly hatched pigeons recruited from hundreds of breeders in the spring and raised in a uniform manner for approximately half a year until the championship in the fall. All pigeons selected in the sample set had the same wild-type feather color (blue) to exclude the potential influence of other pigmentation genes on the iris color. No more than two pigeons from the same breeder were used to ensure unrelatedness among individuals. Pigeons were sampled at approximately five to six months old, when the iris color became stable. The iris color phenotype of each pigeon was examined visually during sampling and photographed. Four to six feathers with follicles were plucked from each individual. A total of 292 pigeons (146 with gravel eyes and 146 pearl eyes) from 237 breeders were gathered for the study. In addition, tissues of the brain, iris, retina, muscle, heart, liver, feather bud, and skin were collected from 14 domestic pigeons (nine with gravel eyes and five with pearl eyes). Tissue samples were

immediately submerged in RNAlater reagent (Qiagen, Germany), stored in RNase-free 5 ml Eppendorf tubes at 4°C overnight, and then transferred to -80°C for long-term storage.

## DNA extraction

Genomic DNA from feather samples was extracted using a DNeasy Blood and Tissue Kit (QIAGEN, Valencia, California, USA) following the manufacturer's instructions. DNA quantity and quality were examined using agarose gel electrophoresis, a NanoDrop spectrophotometer (Thermo Fisher Scientific, USA), and a Qubit fluorometer.

## RNA extraction

RNA was extracted from the brain, iris, retina, muscle, heart, liver, skin, and feather buds of nine gravel-eyed pigeons and the irises of five pearl-eyed pigeons. The tissues were carefully removed from RNAlater and homogenized in TRIzol Reagent (Invitrogen, USA). RNA was isolated following the manufacturer's instructions and stored at -80°C for further use. The absence of RNA degradation and possible contamination was confirmed on 1% agarose gels.

## Whole-genome resequencing, read mapping, and SNP calling

A total of 92 pigeons (49 with gravel eyes and 43 with pearl eyes) from 88 breeders were selected for whole-genome sequencing (S1 Table). Whole-genome resequencing was conducted at Mega Genomics Corporation, Beijing. For each genomic DNA extract, multiplex library preparation with a unique 6-bp sequence index tag was performed following the standard Illumina library construction protocol (Illumina, San Diego, California, USA). The libraries with an average insert size of 250–300 bp were sequenced using an Illumina NovaSeq sequencer, which generated 150 bp paired-end reads, reached an average sequencing depth of 16-fold coverage, and produced an average of 16 Gb of raw sequencing data per individual (S1 Table).

The adaptor sequences at both ends of the reads and bases with Phred quality <30 were trimmed with Cutadapt v1.16 [50]. The processed reads were subsequently aligned to the domestic pigeon reference genome (Cliv_2.1 pigeon genome assembly) with Burrows-Wheeler Aligner v0.7.17 with the default options and parameters [51].

Sequence Alignment/Map (SAM) format files were imported to SAMtools v1.7 for binary format conversion (SAM to BAM) and sorted by coordinates using the default options and parameters [52,53]. We then masked and removed optical or PCR duplicate reads, QC failure reads, unmapped reads, supplementary alignment reads, and nonprimary aligned reads using SAMtools v1.7 such that only the unique mapped reads were retained (S1 Table) [52,53].

SNP and small indel calling was performed in GATK v4.0.2.1 according to the GATK best practices manual with the default parameters [54]. Variant calling was performed with hard filters in GATK v4.0.2.1 and BCFtools v1.3.1 based on these filterExpression parameters in the VariantFiltration algorithm: FisherStrand (FS) >0.3, StrandOddsRatio (SOR) >2.0, RMSMappingQuality (MQ) <50.0, ReadPosRankSumTest (ReadPosRankSum) < -0.05 [52,54].

## LD estimation

Linkage disequilibrium (LD) decay was measured by correlation coefficients ($r^2$) in PopLDdecay v3.40 (http://github.com/BGI-shenzhen/PopLDdecay, accessed 21 Dec. 2018) with the following parameters: -MAF 0.1, -MaxDist 600, -Miss 0.6. The LD decay was plotted as pairwise LD versus pairwise distance between SNPs with a maximum distance of 50 kb using PopLDdecay [55].

## Gene mapping by genome-wide association study (GWAS)

SNPs and indels were filtered with an overall quality score (QUAL) greater than 20, a minor allele frequency (maf) greater than 0.05, maximum missing genotype rates per variant (geno) greater than 0.1, and maximum missing genotype rates per sample (mind) greater than 0.1. The resulting 2,490,056 SNPs from 92 pigeons were used for a genome-wide association analysis (GWAS) with PLINK v1.9 [56], including 49 wild-type individuals set as the control group and 43 pearl-eyed individuals as the case group. The chi-square test was applied for differences between the case and control allele frequency distributions, and the level of significance cutoff was set at $2.01 \times 10^{-8}$ after Bonferroni correction. The Manhattan plot and QQ plot were plotted using the qqman package in R [57].

## Identification of causative mutation

The genotypes of all SNPs with an MAF greater than 0.1 and a P-value less than $2.01 \times 10^{-8}$ in the genomic regions with significant GWAS signals were examined. A region with continuous homozygous genotypes shared by pearl-iris pigeons was considered a candidate region, and the genes within the region were considered candidate genes. SNPs and indels from the candidate region were screened for putative mutations associated with the gravel/pearl iris phenotype. After excluding the SNPs and indels in the noncoding region, SNPs and indels leading to amino acid changes were examined for evolutionary constraints at each affected residue site. The nonsynonymous substitutions at conserved sites among reptile and avian species, or indels causing reading frame shifting or affecting conserved amino acid residues, were considered putative mutations.

## Causal mutation validation

The putative causal *SLC2A11B* mutation was validated in an extended collection of unrelated domestic pigeons with confirmed iris color phenotypes. The sample set consisted of 146 gravel-iris and 146 pearl-iris pigeons from China, including the 92 abovementioned individuals used in GWAS analysis. Full coding exons of *SLC2A11B* were further sequenced in five pearl-iris individuals. The primer sets used to amplify *SLC2A11B* exons (S7 Table) were designed on the basis of the domestic pigeon genome assembly (Cliv_2.1). PCR, subsequent Sanger sequencing, and sequence analysis were performed following previously described procedures [58].

## Transmembrane model prediction

A 2D transmembrane model of SLC2A11B was constructed according to a schematic representation of the GLUT family of proteins from a previous study [59]. The transmembrane regions and orientation of SLC2A11B were predicted by TMpred (https://embnet.vital-it.ch/software/TMPRED_form.html) and TMHMM Server v.2.0 (http://www.cbs.dtu.dk/services/TMHMM/) [60–62].

## Transcriptome sequencing

Transcriptome sequencing of RNA extracts from the iris tissues of four gravel- and five pearl-eyed pigeons was conducted at Novogene Corporation, Beijing, China. RNA quality and purity were evaluated using a NanoPhotometer spectrophotometer (IMPLEN, CA, USA). RNA concentration was measured using a Qubit RNA Assay Kit in a Qubit 2.0 Fluorometer (Life Technologies, CA, USA). The integrity of the RNA was assessed with an RNA Nano 6000 Assay Kit on the Agilent Bioanalyzer 2100 system (Agilent Technologies, CA, USA).

A total of 1.5 µg RNA per sample was used for RNA library preparation. Sequencing libraries were generated using the NEBNext Ultra RNA Library Prep Kit for Illumina (NEB, USA) following the manufacturer's recommendations, and index codes were added to attribute sequences to each sample. Library quality was assessed on the Agilent Bioanalyzer 2100 system. The libraries were sequenced on an Illumina HiSeqXten platform and 150 bp paired-end reads were generated, producing approximately 10 Gb clean data per sample.

## RNA-seq bioinformatics analysis

RNA-seq data were processed in one batch including case (five pearl irises) and control (four gravel irises) individuals. The reads were mapped to the pigeon reference genome assembly (Cliv_2.1) using HISAT v2.1.0 and counted against the predicted gene models using HTSeq-count [63,64]. The total number of aligned reads was normalized by gene length and sequencing depth for an accurate estimation of the expression level. These normalized read counts (TPM and FPKM) were used to represent the expression level of each gene, and differentially expressed genes were determined by DESeq2 [65]. The genes were sorted according to their log2-transformed fold-change values in DESeq2, and a hierarchical clustering algorithm in the pheatmap R package was applied to generate the expression profiles of differentially expressed genes (DEGs) [66]. The absolute $log_2$(fold change) of 1 and padj of 0.05 were set as the threshold for significant DEGs. Analyses of Gene Ontology and KEGG pathway enrichment for 337 DEGs were performed in DAVID v6.8 (S8 Table) [67,68]. The linkage group information for each DEG was extracted from a previous study [34].

## Quantitative real-time PCR (qPCR) validation

Quantitative PCR was performed to determine the *SLC2A11B* expression profiles in the brain, iris, retina, muscle, heart, liver, feather bud, and skin tissues from nine wild-type (gravel eye) pigeons and to validate the differential expression of *SLC2A11B*, *CSF1R*, and *GCH1* between five pearl and four gravel iris tissues. RNA was reverse-transcribed to cDNA using a High-Capacity cDNA Reverse Transcription Kit (Thermo Fisher Scientific, USA) according to the manufacturer's protocol. cDNA was amplified using intron-spanning primers (S7 Table) for each target by quantitative real-time PCR and PowerUp SYBR Green Master Mix (Applied Biosystems, USA) on a QuantStudio 3 Real-Time PCR instrument (Applied Biosystems, USA). Three replicates from each sample were performed to determine the mean value. Beta-actin (*Actb*) was chosen as the internal reference for gene normalization. Experimental data were manually analyzed in a normalized expression comparative Ct ($2^{-\Delta\Delta CT}$) model [69]. The Wilcoxon rank sum test was used to compare the results and differences in expression levels were considered statistically significant if $P < 0.05$.

## Haplotype inference

We downloaded published whole-genome sequencing data for 35 fancy pigeons, two feral pigeons, 10 racing pigeons, and one hill pigeon (*Columba rupestris*) (S4 Table) from NCBI and combined them with our genome data for 92 pigeons to fine map the peal iris causal genes and mutations. VCF files containing the genotypes of scaffold AKCR02000030.1 for all 140 individuals were phased using SHAPEIT v2, with the following parameters:—burn 10—prune 10—main 20—states 200—window 0.1—rho 0.001—effective-size 20000—thread 70 [70]. The phased haplotypes of the *Tr* locus were divided into two clusters: the mutant haplogroup $tr^{mut}$ containing the nonsense mutation (W49X) and the wild-type haplogroup $Tr^+$. The phased file was converted to fasta format and then used for summary statistics and phylogenetic analysis.

## Genetic diversity and selection analysis

The integrated haplotype score (iHS) analysis was applied to the haplotypes spanning the entire AKCR02000030.1 scaffold, to evaluate the extent of excess homozygosity around the ancestral or derived allele [71]. Nucleotide diversity around the *Tr* locus ($\pi$, the average pairwise differences) and Tajima's D (a measure of the skew in the site frequency spectrum) were calculated in 10 kb windows with a 1 kb step size [72], with different combinations of sequence type, population, and $Tr^+$ or $tr^{mut}$ haplotypes. The extended haplotype homozygosity (EHH) score was implemented to validate whether a partial selective sweep had occurred at the $Tr^+$ and $tr^{mut}$ haplotypes. EHH measures the relationship between the frequency of an allele of interest and the amount of LD in the surrounding region and provides the probability that two randomly chosen chromosomes out of a population are homozygous between the core haplotype and the increasingly distant SNP [73]. Once a focal marker was given, the $tr^{mut}$ mutation in this case, the LD decay from the core haplotype was measured for increasingly distant SNPs. The genetic diversity and Tajima's D calculations were performed in TASSEL v5.0 [74], and the significance of differences was tested using the Wilcoxon rank sum test with continuity correction in R. EHH and iHS tests were implemented with the rehh package in R [75].

## Phylogenetic analysis

Phylogenetic trees were built based on $tr^{mut}$ and $Tr^+$ haplotypes across the *SLC2A11B* genic region. Individuals with GQ values less than 5 in the *Tr* mutation site were removed, and a sample set of 140 pigeons consisting of 102 racing pigeons, 35 fancy pigeons, two feral pigeons, and one hill pigeon (*Columba rupestris*) was retained. A 9-kb nonrecombinant region was selected after visual examination, and 24 $tr^{mut}$ and 57 $Tr^+$ haplotypes were identified from the dataset. A minimum evolution (ME) phylogenetic tree was constructed from the 81 haplotypes using the Kimura 2-parameter model and neighbor-joining (NJ) approach as implemented in PAUP v4.0a [76]. The reliability of the nodes in the NJ tree was assessed by 1,000 bootstrap iterations. *C. rupestris* (NCBI: SAMN01057534) was selected as the outgroup. Phylogenetic trees were illustrated with the FIGTREE v1.3.1 package and modified manually.

A median-joining haplotype network was built based on *SLC2A11B* haplotypes from 102 racing pigeons, 35 fancy pigeons, and two feral pigeons. The haplotypes were reformatted using DnaSP v6.12.03 [77] and constructed into a network using the network approach in PopART software [78].

## TMRCA estimation of the $tr^{mut}$ haplotypes

To trace the origin of the pearl iris mutation in the domestic pigeon, the most recent common ancestor (TMRCA) for all $tr^{mut}$ alleles was estimated using startmrca [79]. which leverages both the recombination rates and the accumulation of new mutations of the targeted allele's ancestral haplotype. Relative to other approximate Bayesian computation methods, this approach is based on a hidden Markov model and the assumption that the focal allele is subject to positive selection. Individuals homozygous for the $Tr^+$ allele were set as the reference panel. The selected and reference panels were set at 100 and 40 for each run. To obtain selection-onset time, independent Monte Carlo Markov (MCMC) chains were run 10 times, each with 200,000 iterations (first 6,000 iterations discarded as burn-in). The result with the highest posterior probability was considered the TMRCA estimate. To obtain confidence intervals, we took the 2.5th and 97.5th quantiles of each resulting distribution and calculated the recombination rates of the scaffold AKCR02000030.1 using FastEPRR with nonoverlapping 10 kb and 50 kb window sizes [80]. We used a mutation rate of $1.42 \times 10^{-9}$ per site per generation [21] and a generation time of one year.

## Identification of *SLC2A11B* variation in Aves

We downloaded genome data for 34 avian species whose iris pigments were documented from NCBI (S5 Table) and extracted *SLC2A11B* coding sequences with blastn v2.7.1 [81]. *SLC2A11B* orthologs from two Crocodilia (*Crocodylus porosus*, *Alligator mississippiensis*), six Testudines (*Pelodiscus sinensis*, *Chelonoidis abingdonii*, *Chrysemys picta bellii*, *Terrapene Carolina triunguis*, *Pelusios castaneus*, *Gopherus agassizii*), and nine Squamata (*Anolis carolinensis*, *Varanus komodoensis*, *Pogona vitticeps*, *Salvator merianae*, *Naja naja*, *Pseudonaja textilis*, *Pantherophis guttatus*, *Laticauda laticaudata*, *Notechis scutatus*) were downloaded from the Ensembl database (http://asia.ensembl.org/index.html, Release 101). These 51 coding sequences were aligned using MUSCLE (codon) in MEGA v10.1.8 [82] for variant detection. The potential impacts of missense variations at evolutionarily conserved amino acid sites on protein function were predicted with SIFT [83].

## Detection of relaxed selection in the avian *SLC2A11B*

To test for relaxed or intensified selection of *SLC2A11B*, we selected 41 orthologs of the zebrafish *SLC2A11B* and *TYR* genes in the Ensembl database (http://asia.ensembl.org/index.html, Release 101) and aligned the CDSs according to the corresponding amino acid sequence with the L-INS-i algorithm in MAFFT v7.471 [84]. The CDSs of two cormorant species were truncated so that only the 1,434 bp before the nonsense substitution were retained. To obtain the tree topology of 41 vertebrate species, the initial tree was generated by http://timetree.org/and modified according to the species tree topology reported in various phylogenetic studies [85–88]. *TYR*, the key melanogenesis gene that is expected to be evolutionarily conserved and under consistent selective pressure across vertebrates, was used as a control in the analysis.

The RELAX test implemented in the HYPHY package [42] takes some branches in the tree as test branches and some others as reference branches (there can be unclassified branches) and infers a parameter $k$, which is the selection strength of positive or negative selection (i.e., deviation of omega from 1) on the test branches divided by that on the reference branches. Hence, if $k$ is significantly smaller than 1, the selection on test branches is relaxed relative to that on reference branches; if $k$ is significantly larger than 1, intensified selection is suggested. The level of significance of $k$ is tested by a likelihood ratio test (LRT). Three different schemes of test/reference branch designation (basal, internal and clade) were applied: (1) the basal scheme assigns only the basal branches of each monophyletic group as test/reference branches; (2) the internal scheme assigns all internal branches of a monophyletic group as test/reference branches; and (3) the clade scheme assigns all (internal and external) branches within a clade as test/reference branches.

The 41 taxa involved in this analysis belong to five major monophyletic groups: Aves (birds), Crocodilia (crocodiles), Testudines (turtles), Squamata (scaled reptiles), and Teleostei (ray-finned fishes). For each scheme, we conducted the RELAX test separately for each group, with branches of each group used as test branches against corresponding branches from the other four groups as reference branches. We performed identical tests for the CDS of *SLC2A11B* and the negative control, *TYR*.

## Supporting information

**S1 Fig. The linkage disequilibrium (LD) decay in the pigeon genome.** The LD decay curve is based on the mean correlation coefficient ($r^2$) between common SNPs (minor allele frequency $\geq$ 0.1). The threshold for "useful LD" is set with $r^2 < 0.2$ at distances beyond 0.9 Kb. Under this scenario, the pigeon genome shows a rapid LD decay suggesting that the genome-wide SNPs generated from the sample set are nearly or completely independent from each

other, and hence are sufficient for association mapping in pigeons.
(TIF)

**S2 Fig. Quantile-Quantile (QQ) plot for GWAS.** The observed versus expected quantiles of the genome-wide association *P*-value shown in Fig 2A.
(TIF)

**S3 Fig. Prediction of transmembrane regions and orientation of SLC2A11B protein based on the whole SLC2A11B sequence of 496 amino acids.** (A) TMHMM posterior probabilities of inside/outside/TM helix. The N-best prediction is displayed at the top where transmembrane regions are shown in red boxes. (B) Result output from TMpred server. The predicted transmembrane helices with scores above 500 are considered significant. The solid and dashed line indicates inside-to-outside and outside-to-inside transmembrane helices, respectively.
(TIF)

**S4 Fig. Gene expression profile between pearl and gravel irises by RNA-seq.** The heatmap, hierarchically clustered into two groups of genes, recapitulates a total of 337 differentially expressed genes (DEGs), among which 295 and 42 genes were specifically upregulated in gravel (N = 4) and pearl (N = 5) irises, respectively. Columns are individual samples and rows indicate individual genes. The level of expression is color-coded from DESeq normalized counts, with red representing the higher level of expression, blue the lower level.
(TIF)

**S5 Fig. Functional analysis of differentially expressed genes (DEGs) based on RNA-seq data.** (A) GO enrichment of 295 DEGs upregulated in gravel iris. (B) GO enrichment of 42 DEGs upregulated in pearl iris. The bubble diagrams show the degree of enrichment of Gene Ontology (GO) terms in three categories. The orange, blue, and black represent molecular function (MF), cellular component (CC), and biology process (BP) categories, respectively. Each bubble indicates a GO term, and the size of bubbles is proportional to the number of genes annotated to the GO term. P-value is represented by the color map.
(TIF)

**S6 Fig. Haplotype network of 55 *Tr*⁺ and 24 *tr*ᵐᵘᵗ haplotypes.** The haplotype network was generated from 9 Kb nonrecombining *Tr* region from 139 domestic pigeons (35 fancy pigeons, 2 feral pigeons, and 102 racing pigeons). The mutations are shown by hatch marks. The orange and blue circles represent *tr*ᵐᵘᵗ and *Tr*⁺ haplotypes, respectively.
(TIF)

**S7 Fig. Selection analysis of haplotypes containing W49X mutation (*tr*ᵐᵘᵗ) and wild-type haplotypes (*Tr*⁺) surrounding the *Tr* locus.** The integrated haplotype score (iHS) was calculated for scaffold AKCR02000030.1 in fancy pigeons (A) and racing pigeons (B). The gray lines represent the significance level of absolute iHS scores of 3 or greater. The extended haplotype homozygosity (EHH) decay across the *Tr* locus region is showed for fancy pigeons (C) and racing pigeons (D).
(TIF)

**S8 Fig. The nonsense, missense and frame-shifting mutations of SLC2A11B in avian species.** A total of 52 coding sequences of 35 Aves, 2 Crocodilia, 6 Testudines, and 9 Squamata were aligned. The partial alignment of amino acid sequences of avian SLC2A11B with conserved species-specific mutations is shown. The conserved species-specific nonsense, missense and frame-shifting mutations are labeled on the top of the table, and the deleterious mutations

are marked in red.
(TIF)

**S1 Table. Sample information and resequencing data statistics.**
(XLSX)

**S2 Table. Sample information and RNA-seq data statistics.**
(XLSX)

**S3 Table. Differentially expressed genes from RNA-seq.**
(XLSX)

**S4 Table. Sample information and resequencing data statistics from NCBI SRA library involved in this study.**
(XLSX)

**S5 Table. Thirty-four avian genomes used for alignment of *SLC2A11B* coding sequence.**
(XLSX)

**S6 Table. RELAX test results for *SLC2A11B* and *TYR*.**
(XLSX)

**S7 Table. Primer sequences used in this study.**
(XLSX)

**S8 Table. GO and KEGG pathway enrichment for differentially expressed genes.**
(XLSX)

## Acknowledgments

We thank J. -O. Gao and his team for coordinating the sampling logistics, H. Meng, X. Sun, H. Yu, Y. -C. Liu, Y. -T. Xing for collecting samples, C. Xie for helpful discussion, K. -W. Jiang for providing avian samples for validation, and all the pigeon owners for donating samples.

## Author Contributions

**Conceptualization:** Xiao Xu.

**Data curation:** Xiao Xu.

**Formal analysis:** Si Si, Xiao Xu.

**Funding acquisition:** Xiao Xu, Shu-Jin Luo.

**Investigation:** Si Si, Xiao Xu, Yan Zhuang, Zhengting Zou.

**Methodology:** Xiao Xu, Yan Zhuang, Zhengting Zou.

**Project administration:** Xiao Xu.

**Resources:** Xiaodong Gao, Honghai Zhang, Shu-Jin Luo.

**Software:** Si Si, Zhengting Zou.

**Supervision:** Xiao Xu, Shu-Jin Luo.

**Validation:** Si Si, Xiao Xu.

**Visualization:** Si Si, Zhengting Zou.

**Writing – original draft:** Si Si, Xiao Xu.

**Writing – review & editing:** Si Si, Xiao Xu, Zhengting Zou, Shu-Jin Luo.

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
