## [Decision Letter · Decision Letter 0]

10 May 2021

Dear Dr Xu,

Thank you very much for submitting your Research Article entitled 'The genetics and evolution of eye color in domestic pigeons (Columba livia)' to PLOS Genetics.

The manuscript was fully evaluated at the editorial level and by independent peer reviewers. The two reviewers thought the paper was interesting and on an important problem, but raised some concerns about the current manuscript. Based on the reviews, we will not be able to accept this version of the manuscript, but we would be willing to review a revised version. We cannot, of course, promise publication at that time. In your revision, please be make sure that your results are consistent with your data (e.g. gene expression of SLC2A11B, RELAX results of TYR), include statistical analyses wherever appropriate (e.g. gene expression comparisons), and avoid over-speculation in the interpretation of your data (e.g. the mechanism of SLC2A11B). The reviews are both very thoughtful and thorough, and have many additional suggestions for how you can improve your paper.

Should you decide to revise the manuscript for further consideration here, your revisions should address the specific points made by each reviewer. We will also require a detailed list of your responses to the review comments and a description of the changes you have made in the manuscript. Please include line numbers in your revision.

If you decide to revise the manuscript for further consideration at PLOS Genetics, please aim to resubmit within the next 60 days, unless it will take extra time to address the concerns of the reviewers, in which case we would appreciate an expected resubmission date by email to plosgenetics@plos.org.

[LINK]

We are sorry that we cannot be more positive about your manuscript at this stage. Please do not hesitate to contact us if you have any concerns or questions.

Yours sincerely,

Kelly A. Dyer

Associate Editor

PLOS Genetics

Bret Payseur

Section Editor: Evolution

PLOS Genetics

Reviewer's Responses to Questions

**Comments to the Authors:**

Reviewer #1: Uploaded as an attachment.

Reviewer #2: In this study the authors identify genetic variants potentially involved in the pearl eyed phenotype of domestic pigeons. After successfully demonstrating a causative variant, the authors search for associations with eye color in other birds and also test the hypothesis that this gene has undergone relaxed selection in birds. I found the paper to be well written with many interesting observations carefully unpacked. I read the paper with consideration of another similar publication in the same journal by Andrade et al, which also identified the same causative variants. I applaud the authors of the current study for their phylogenetic comparative analyses, which goes beyond what was done in the other manuscript, and is a compelling addition.

I enjoyed the analyses conducted on the gene of interest with other avian species. I expect these types of analyses will become more common with the many genomes becoming available and I see this as a good example of how fruitful these comparative analyses are going to be. On the other hand, I did not find the analyses conducted on the history of the gene outside of birds to be as easy to follow. That birds have retained some sort of functional expression of SLC2A11B in the eye, but not other tissue, is certainly interesting. However, I found it difficult to 1) postulate that this gene is degenerate when it is associated with eye color across numerous birds, 2) that a relaxation of selection on this gene is due to the evolution of feathers in birds when many non-avian dinosaurs had feathers, 3) link the data in the paper about eye coloration to a broad phylogenetic analysis that is largely based on dermal pigments, and 4) fully embrace the results when TYR shows inconsistent result with what is written in the paper. I expand on this somewhat in my comments below, but I personally would gain the same value from this manuscript if the RELAX analysis were not included and the focus was more specifically on the role of SLC2A11B in birds in general. An enlightening discussion could still be made more broadly about the history of chromatophores in the reptilian radiation.

The version of the ms I received did not have line numbers, so I do my best to be specific with pages and paragraphs in my comments below:

Abstract: “elucidate” is vague as to what will be made clearer. Reflecting on what is done in the study, I suggest “explore” as a potential alternative here.

Page 3, paragraph 2: last sentence – birds are vertebrates so this should possibly read “other vertebrates” or be expanded.

Paragraph 3 –Intraspecific variation in wild birds is often associated with age (e.g. https://doi.org/10.1111/j.1474-919X.2006.00630.x) and sex (examples include accipiters, bushtits, others). Suffice to say intraspecific variation is not limited to domestic species.

Page 4, 2nd paragraph: A citation is needed for the types of pigment cells in other avian species

Figure 1 should have a photo credit

Page 4, last paragraph: “aves coloration” should be “avian coloration”

Page 6 “The iris pigmentation in pearl-eyed pigeons resembles that of SLC2A11B loss of function in fish, supporting W49X of SLC2A11B as the causal mutation responsible for the pearl eye color” In what way is the iris pigmentation similar, in the pigments themselves or in appearance?

Page 6: “...mutation was further validated in..” By which method were they validated?

Page 6: “However, five pearl-eyed pigeons were heterozygous for the SLC2A11B p.W49X variant (Fig. 3B) with no other mutations found in the exons of SLC2A11B.” For ease of readership, I suggest that there is a mention of the argument described in discussion that these could have been mis-phenotyped.

Page 7/Figure 4A: Be more specific about high or low expression (it is only meaningful in relation to other tissue), I suggest including a statistical test that will allow you to state that SLC2A11B expression is higher in brain, iris, retina, and muscle than other tissue.

Page 7: One alternative (although not mutually exclusive) explanation for the results in Andrade et al is that there were two other genes also on the same linkage group that were found to be differentially expressed (CHEK2 and TRAFD1). The potential for other cis-regulatory variants located in the region of differentiation detected here is unlikely, but possible. It is worth checking these two in the RNAseq data as it will support, or perhaps suggest against, this possibility of other regulatory variants.

Page 7: “SLC2A11B was slightly downregulated, by 1.2- fold, in the pearl iris (Fig. 4B), likely caused by nonsense-mediated RNA decay.” I found it disingenuous to not point out that SLC2A11B is not significantly differentially expressed in the RNAseq analysis in this place (it is only in the supplemental that I can make this connection). What is the explanation for the significant difference in qPCR, but not RNAseq? It is difficult to speculate based on Figure 4C, which would benefit from including an overlay of the data points on the boxplots.

Page 8: “Such widespread prevalence across pigeon breeds is consistent with a relatively longstanding history of pearl eye color in domestic pigeons.” This could be interpreted in one of two ways – that the genotype is old, or that the pearl eye phenotype is widespread across breeds. My suggestion is that if it is the latter, to put a number on how many of these 36 breeds are known to have the pearl eyed phenotype (or if it is all breeds, write that).

Page 8: As this is the first mention of C. rupestris, I suggest you make a small note that this species is considered sister to all others in the dataset.

Page 8: Figure 5 caption does not match the text. The text reads that a NJ tree was made from the SLC2A11B genic sequence, but the caption reads that it was the 9kb region of association. I do not know which is shown in the figure.

Page 8: What method was used to date the TMRCA?

Page 9: The section title would be more appropriately written as “SLC2A11B is associated with avian iris color variation”

Page 11: “On the other hand, the absence of relaxed selection was consistently supported by all three schemes for TYR, a key melanogenesis gene that is expected to be evolutionarily conserved and under similar selective pressure across all vertebrates.” This is at odds with the table in Figure 7 – here k <1 and p = 5e-9 for TYR in the clade comparison.

Page 12: “as most likely responsible” would be appropriate in this case

Page 13: small typo: “it might be more appropriate to refer *to* pigeon iris stromal cells”

Page 13: “which indicated that CSF1R acts downstream of SLC2A11B in the regulatory network” this is somewhat speculative, I suggest some caveats such as “which is consistent with the possibility that CSF1R is downstream...” or similar

Page 14: “One landmark evolutionary transition setting the stage for the divergence of avians from their dinosaur ancestors was the development of feathery coverage that concealed their skin pigmentation.” This is an oversimplification given that there are now numerous fossilized non-avian dinosaurs suggesting that some dinosaurs had feathers. It is also an oversimplification to state that the only important pigments in birds are related to feather color, as many species have colorful bare parts (e.g. see this review on bare parts signaling https://doi.org/10.1642/AUK-16-136.1). Although this tells a compelling story arc, the inconsistency in results relating to TYR in the analysis, and the uncertainty of SLC2A11B’s role in other coloration types in birds, would give me some uncertainty in describing the history of this gene in this way. I also find it difficult to reconcile the widespread phylogenetic signal of SLC2A11B’s role in eye coloration across birds (suggesting at least some degree of purifying selection is taking place) and the hypothesis that this gene has undergone evolutionary demise.

“dinosaur ancestors” could be rephrased as non-avian dinosaur.

Page 15: The IACUC # should be appropriate here

Page 17: It is not clear which versions of GATK were used for what steps during SNP calling

Page 19: What metric was used to determine significantly differentially expressed genes in DESeq2? E.g. a p-value cutoff, or p-value + log fold change cut off.

Page 20: This should have a citation or perhaps expand on the methods used “Experimental data were manually analyzed in a normalized expression comparative Ct (-2∆∆Ct) model.”

Supplemental information:

Figure S1: “Despite the rapid LD decay indicated that the clumps of SNPs were nearly or completely independent from each other, domestic pigeons were well suited for association-mapping methods considering on the sufficiently genome- wide marker density generated by SNPs calling.” This should be rephrased for clarity.

I believe some of the genomes in Table S5 (e.g. Anna’s Hummingbird) should be attributed to this preprint from VGP: https://doi.org/10.1101/2020.05.22.110833

**Have all data underlying the figures and results presented in the manuscript been provided?**

Reviewer #1: Yes

Reviewer #2: Yes

PLOS authors have the option to publish the peer review history of their article (what does this mean?). If published, this will include your full peer review and any attached files.

Reviewer #1: No

Reviewer #2: No

---

## [Decision Letter · Decision Letter 1]

12 Jul 2021

Dear Dr Xu,

Thank you very much for submitting your Research Article entitled 'The genetics and evolution of eye color in domestic pigeons (Columba livia)' to PLOS Genetics.

The manuscript was fully evaluated at the editorial level and by independent peer reviewers. The reviewers appreciated the attention to an important topic but identified some concerns that we ask you address in a revised manuscript.  In particular, please address the reviewer's concern about your interpretations of the RELAX analyses.  Please also carefully copy edit your manuscript, as there were several typos inserted in the places where revisions occurred. 

You mention in your cover letter that there was a paper on a similar topic that was published recently in PLoS Genetics. Your paper was reviewed as part of our "scooping policy" (https://journals.plos.org/plosgenetics/s/journal-information#loc-criteria-for-publication), and the reviewers were made aware of both this policy and previously published paper when we invited them to review your paper.

We therefore ask you to modify the manuscript according to the review recommendations. Your revisions should address the specific points made by each reviewer.

[LINK]

Yours sincerely,

Kelly A. Dyer

Associate Editor

PLOS Genetics

Bret Payseur

Section Editor: Evolution

PLOS Genetics

Reviewer's Responses to Questions

**Comments to the Authors:**

Reviewer #1: The changes to the sections

"Genes with known functions in xanthophore/leucophore pigmentation are downregulated in SLC2A11B W49X pigeon iris" and "SLC2A11B is associated with avian iris color variation" address all the major points I raised in my original review. It was a pleasure to review.

Reviewer #2: I was reviewer #2 on the original submission of the manuscript. I appreciate the responses to my comments on the earlier version, and feel that the manuscript has improved in this revision.

I spent quite a bit of time revisiting the comments regarding the RELAX analysis, and re-reading the original manuscript for the software (Wertheim et al 2015). I appreciate that the authors gave my comments some thought, which has helped guide me on how the authors are interpreting the data regarding evolutionary demise. One point I remain unclear on is why the test is only considered significant when there are all three schemes supporting a model of relaxed selection. There seems to be compelling evidence that TYR is under relaxed selection in birds and the same can be said for the relaxed selection in squamata and the testudines, which also suggest relaxation under some schemes (table S6).

Some further justification, when these schemes are introduced in the results, would help clarify the interpretation that SLC2A11B has undergone relaxed selection. For example, is there a reason the authors believe one scheme over another? I imagine the internal nodes may have lower sampling variance as there is more data available than on single branches (as seems to be indicated in the RELAX manuscript), but in that case it is not clear why setting up different schemes is necessary vs. reporting values of k at nodes and terminal branches.

I appreciate the compelling premise that dermal chromatophores have undergone demise, a relaxation of selection has occurred for chromatophore-linked genes such as this, and then possibly a chromatophores have been repurposed in the eyes of birds. However, as there is some speculation involved here, it is key is that the molecular data backs this up as the most likely explanation, or at least there is clear justification for the way the data is presented and interpreted.

**Have all data underlying the figures and results presented in the manuscript been provided?**

Reviewer #1: Yes

Reviewer #2: Yes

PLOS authors have the option to publish the peer review history of their article (what does this mean?). If published, this will include your full peer review and any attached files.

Reviewer #1: No

Reviewer #2: No

---

## [Editor Report · Decision Letter 2]

10 Aug 2021

Dear Dr Xu,

We are pleased to inform you that your manuscript entitled "The genetics and evolution of eye color in domestic pigeons (Columba livia)" has been editorially accepted for publication in PLOS Genetics. Congratulations!

Yours sincerely,

Kelly A. Dyer

Associate Editor

PLOS Genetics

Bret Payseur

Section Editor: Evolution

PLOS Genetics

Comments from the reviewers (if applicable):

**Data Deposition**

http://datadryad.org/submit?journalID=pgenetics&manu=PGENETICS-D-21-00207R2

**Press Queries**

---

## [Editor Report · Acceptance letter]

24 Aug 2021

PGENETICS-D-21-00207R2 

The genetics and evolution of eye color in domestic pigeons (Columba livia) 

Dear Dr Xu, 

We are pleased to inform you that your manuscript entitled "The genetics and evolution of eye color in domestic pigeons (Columba livia)" has been formally accepted for publication in PLOS Genetics! Your manuscript is now with our production department and you will be notified of the publication date in due course.

With kind regards,

Andrea Szabo

PLOS Genetics

On behalf of:
